# Influence of the Characteristics of Sports Sharing Economy Services on Intention of Use: The Mediating Effect of Consumer Attitude and Trust

**Sunjung Kim [1] and Kyongmin Lee [2,*]**

1   The Educational Research Institute, Sungshin Women's University, Seoul 02844, Korea; kimsj@sungshin.ac.kr
2   Department of Sport Industry, Korea National Sport University, Seoul 05541, Korea
*   Correspondence: dsukmlee77@outlook.kr

**Abstract:** This study empirically investigated the effectiveness of the characteristics of sports sharing economy services that has been highlighted recently. The study used consumers' attitude and trust as parameters of the association between the characteristics and use intention of consumers. There are three research questions in the study. The first question is whether the characteristics of sports sharing economy services affect the consumers' use intention. The second question is whether the characteristics of sports sharing economy services affect the attitude of consumers and the trust level in sports sharing economy services. The third question is whether the attitude and trust level of consumers mediate sports sharing economy services and use intention. A survey was conducted among college students using a convenient sampling method. We found that: (1) service characteristics such as usefulness, bonding, and consistency had a positive influence on use intention, and risk factor had a negative influence; (2) the service characteristics significantly influenced consumer attitudes and trust of the sharing economy services; usefulness, bonding, and consistency had a positive influence, and risk factor had a negative influence; and (3) the attitude and trust of consumers toward sharing economy services affected use intention; usefulness, bonding, and sustainability had a positive influence, while risk factor had a negative influence. Thus, it was confirmed that the attitude and trust level of consumers mediated sports sharing economy services and use intention. This study also suggested academic and practical implications to establish a more effective sports sharing economy service marketing strategy and develop quality content in relation to the service characteristics and consumers.

**Keywords:** sports sharing economy services; use intention; consumer attitude; trust; mediating effect

## 1. Introduction

Following the advent of information and communications technologies (ICT) and the fourth industrial revolution platform, sharing economy services are expanding significantly.

The market size of sharing economy services is expected to grow over 20 times in 2025 compared with the growth registered in 2015, reaching $335B from $15B (KITA 2015). The market size of sharing economy services worldwide has been growing remarkably, at an average of 78% annually since 2010. Airbnb, a leading accommodation sharing service founded in 2007, has grown significantly. For instance, it has an annual revenue of US$29.3B. This exceeds Hilton's annual revenue of US$23.6B, which has traditionally taken first place in the hotel industry. Additionally, Airbnb currently provides more than 3 million accommodations across 65,000 cities and 191 countries (Hur 2020). The market size of sharing economy services in China has also increased to about 40% of the annual growth rate with support from the government. The sharing economy services in South Korea include various domains that are closely related to daily life, such as food (delivery), accommodation, taxi, real estate, beauty, rhythmic exercise, training, fitness, and health (Joo 2018).

The sports sharing economy services market was accelerated by the COVID-19 pandemic, during which its scope was further expanded. The sports sharing economy services are smaller than the sharing economy services; in contrast, this market is considered to have infinite growth potential (Kim 2020; Park 2020). It started in the US and has spread around the world. Lime, which was started in the US in 2017, has grown into a startup company estimated at a corporate value of over US$1B within two years of establishment. Moreover, the company was highlighted by attracting an investment of approximately US$310M in 2019 (Howmuch Team 2020). Reflecting such market demands, a tremendous number of electric kickboards are being operated in a sharing manner in 47 cities and universities across 23 countries, including France, Mexico, Australia, and Singapore. Shared bike services were first introduced in the Netherlands, and Uber, a shared economy service company in the US, was the first to start a shared bike business with the acquisition of "Jump Bikes." The services are mainly used for short-distance travel, fitness, and diet exercises among teenagers and people in their 30s (Yoo 2021). The shared-bike services model in South Korea is different from those in other countries in that there are no designated return parking lots. In South Korea, bikes and electric kickboards operated by the Seoul Metropolitan Government have been designated as the leading sports sharing economy services (Kang 2020). This study's contributions are as follows. First, there are many domestic and foreign studies on sharing economy services; however, there are relatively few studies on sharing economy services, hence the need for a relevant study. Second, especially in South Korea, there are few studies on the characteristics of sports economy services, and almost no studies on the characteristics of sports sharing economy services. Third, it is even more difficult to find studies on the effectiveness of bikes and electric kickboards among the studies on the characteristics of sports sharing economy services.

Research on sharing economy services can be broadly divided into P2P (peer to peer) and B2P (business to peer). The sharing economy service research for P2P has been conducted in the form of sharing personal ownership and deals with themes such as vehicle sharing, lodging sharing, and baby product sharing (Kim 2021; Lim et al. 2020). For B2P-targeted sharing economy service research, Min and Kim (2019) conducted a sharing economy service study in a local tourist area, and Yoon et al. (2017) conducted a bicycle sharing service study. Yoon et al. (2021) conducted a study focusing on the sharing economy services of Chinese companies.

The motive for using the sharing economy service is gradually connected from the company to the individual and the flow of items is diversified (Shim 2019). By linking the characteristics and motivations of the sharing economy service, it can be seen that consumers are increasingly using services that take place at a specific time and space, such as trading used goods, lodging, vehicles, sharing knowledge and experiences, and sharing jobs (Lim et al. 2020). This usage pattern is expanding the range of shared items from high-end items to free sharing, from high prices to low prices (Kim and Kim 2020). Therefore, the motivation for using the sharing economy service can be said to be economic value including diversity obtained through sharing knowledge, experience, and hobbies.

This study notes that there are relatively few studies on the sports domain, especially on the mediation effects between the characteristics of sports services and use intention. However, many studies on sharing economy services are available both domestically and overseas. Thus, this study addressed the following research questions:

RQ1: Does the characteristics of sports sharing economy services affect the consumers' use intention?

RQ2: Does the characteristics of sports sharing economy services affect the attitude of consumers and the trust level in sports sharing economy services?

RQ3: Does the attitude and trust level of consumers mediate sports sharing economy services and use intention?

This study aimed to verify the effectiveness of the attitude and trust of consumers, which are parameters in the relationship between the characteristics of sports services and use intention.

## 2. Literature Review

The sharing economy is a concept first used by Professor Lawrence Lessig at Harvard University in 2008. The sharing economy refers to an economic way in which a product that is produced once is shared by several people. The sharing economy has both the attributes of a market economy and the attributes of gift exchange. With the development of information technology, products and services have been digitized and the sharing economy has become possible (Shin 2014). Additionally, the sharing economy service makes it possible to use goods together instead of owning them based on digitalization and facilitates decentralized peer-to-peer transactions (Shin 2014).

Sharing economy services have five characteristics. First, sharing economy services, as a large-scale market based on the sharing economy, contain the characteristics of potentially higher levels of economic activity, and this economic activity enables the exchange of goods with the advent of new services (Pae et al. 2019). Second, sharing economy services are characterized by high-impact capital. It can lead to new opportunities in all areas, ranging from assets to time and money (Sunwoo and Im 2021). Third, the sharing economy service has the characteristic of being a mass-based network, and this sharing economy service has the characteristic that more individualized capital and labor are supplied rather than based on the state, corporations, or centralized institutions or classes (Lim et al. 2020). Fourth, the sharing economy service has a characteristic that the boundary between individuals and experts is not clear and that the provision of labor and services is commercially traded (Pae et al. 2019). Fifth, another characteristic of the sharing economy service due to the ambiguity of the boundary between work and leisure is that it deviates from the traditional concept of work characterized by full-time employees, commitment, performance pay, economic dependence, and performance pay levels (Sundararajan 2017).

### 2.1. Relationship between Characteristics of Sharing Economy Services and Use Intention

The characteristics of sharing economy services stem from the motivation theory. This theory is used to explain the acceptance of information technology of individuals in the domain of sharing economy services, and classifies motivation as external or internal. In external motivation, behavior is influenced by practical factors such as external rewards, whereas in internal motivation, it is influenced by pleasure, fun, satisfaction, and accomplishment (Möhlmann 2015). The technology acceptance model (TAM) has a significant influence on the attitude and use intention of individuals in cases where individuals accept new information technologies voluntarily, which is suitable for explaining the characteristics of sharing economy services (Venkatesh and Davis 2000). Ryu and Lee (2017) reported that the characteristics of sharing economy services include risks, and that when using these services, the higher the perceived risks, the more the negative influence on the use intention (Kim 2017). Cho and Jeoung (2019) argued that sharing economy services are characterized by usefulness, sustainability, and bonding, all of which are proportional to the use intention of consumers. As a result of examining preceding studies, various studies on sharing economy were conducted. However, in South Korea, a study on the characteristics of sharing economy services in the sports domain is still in its early stages. In addition, no study has been conducted to analyze empirical relations based on the motivation theory and TAM, hence the need for this study. To this end, the following hypothesis was established with a theoretical basis in terms of the synchronous theory and TAM.

**Hypothesis 1 (H1).** *The characteristics of sports sharing economy services influence the use intention.*

### 2.2. Relationship between Characteristics of Sharing Economy Services and Consumer Attitude

The attitude of consumers refers to the identification of their emotional states so that either a positive or negative response can be evaluated (Ryu and Lee 2017), and has long been widely perceived as an important variable to predict social behavior (Ajzen 1991;

Fishbein and Ajzen 2010). Lee (2016), in a study on sharing economy services for college students, argued that the attitude of consumers is an important antecedent variable that can predict of the outcome of their use intention. Hao and Kim (2020) also claimed, in their study on the use intention of sharing bikes services in China, that the perceived usefulness of sports sharing economy services has a positive influence on the attitude of consumers. Additionally, a study conducted by college students showed that the characteristics of unique economic services influence consumer attitudes (Ryu and Lee 2017).

Meanwhile, the sharing services of bikes and electric kickboards are constantly evolving. There are several considerable factors that could potentially influence consumers' use intension for these services, such as the implementation of a face recognition payment, battery protection and management technologies, and a convenient credit payment system. These factors generate different consumer attitudes regarding usefulness, bonding, sustainability, and risk. Therefore, this study established the following hypotheses based on whether the characteristics of sharing economy services influence the attitude of consumers in the sports domain.

**Hypothesis 2 (H2).** *The characteristics of sports sharing economy services influence the attitude of consumers.*

**Hypothesis 2a (H2a).** *The usefulness of sports sharing economy services has a positive influence on the attitude of consumers.*

**Hypothesis 2b (H2b).** *Bonding of sports sharing economy services has a positive influence on the attitude of consumers.*

**Hypothesis 2c (H2c).** *The sustainability of sports sharing economy services has a positive influence on the attitude of consumers.*

**Hypothesis 2d (H2d).** *The risk of sports sharing economy services has a positive influence on the attitude of consumers.*

*2.3. Relationship between Characteristics of Sharing Economy Services and Consumer Trust*

The characteristic factors of sports sharing economy services are expected to influence the consumer–trust relationship. According to a previous study, usefulness serves as a motive for consumers (Deci and Ryan 1985). The more consumers perceive that a specific sharing economy service can achieve an objective efficiently, the more they develop a positive attitude toward the service. Moreover, the use intention for the service will also be higher (Wu et al. 2019). In this study, the more consumers perceive that bikes and electric kickboard sharing services (which can be utilized as a sports sharing economy) allow for movement to places where physical activity is possible but movement is difficult, the higher the reliability of consumers will be. As an extension of the rational behavior theory, the following hypotheses were established with regard to the characteristics and reliability of sports sharing economy services:

**Hypothesis 3 (H3).** *The characteristics of sports sharing economy services influence the trust of consumers.*

**Hypothesis 3a (H3a).** *The usefulness of sports sharing economy services has a positive influence on the trust of consumers.*

**Hypothesis 3b (H3b).** *Bonding of sports sharing economy services has a positive influence on the trust of consumers.*

**Hypothesis 3c (H3c).** *The sustainability of sports sharing economy services has a positive influence on the trust of consumers.*

**Hypothesis 3d (H3d).** *The risk of sports sharing economy services has a positive influence on the trust of consumers.*

*2.4. Relationship between Consumer Attitude and Use Intention*

According to a previous study on the relationship between consumer attitude and use intention of sharing economy services, consumer attitude refers to a consumer's mental will at the moment of purchasing, and implies a probability that consumer attitude is directly converted into actions (Blackwell et al. 2001). Such consumer attitude influences use intention, which is based on the theory of rational behavior by Ajzen and Fishbein (1975). Reliability influences the purchase intention. Therefore, it is very important to instill positive emotions in the brand (Lee et al. 2016a). There are studies that verify consumer attitude using parameters in the characteristics of sharing economy services (Cho 2019). However, in the sports domain, almost no study has used consumer attitude as a parameter. Therefore, the following hypothesis was established based on the rational behavior theory to determine whether a mediating effect exists between consumer attitudes and use intention.

**Hypothesis 4 (H4).** *Consumer attitude toward the sports sharing economy services has a positive influence on the consumers' use intention.*

*2.5. Relationship between Consumer Trust and Use Intention*

Trust is an important factor in socioeconomic interactions in which uncertainty and dependence coexist. Furthermore, it accepts risks and uncertainties to reduce social complexity (Luhmann 2018). Moreover, it is possible to infer that the trust level will be formed by the degree of expectation entailed in a specific situation. In a previous study on the trust and use intention of sharing economy services, Jung and Chung (2007) argued that the factors that have a positive influence on the trust level of consumers who use websites include the convenience and ease of using the websites, while Han and Lee (2016) stated that the trust level of a buyer is influenced by the sense of social connectivity to a supplier in transactions within the B2B market. Ji and Byun (2011) claimed that the trust level in a hotel is associated with the eco-friendly service factors it promotes. Lee (2018) further suggested that the low level of trust in the use of sharing accommodation is attributable to risk perception on social and financial risks as well as safety. The following hypothesis was established to determine whether the trust level of consumers against sharing economy services in the sports domain is a parameter influencing use intention.

**Hypothesis 5 (H5).** *The trust level in sports sharing economy services has a positive influence on the use intention of consumers.*

*2.6. Mediating Effect of Consumer Attitude and Consumer Trust on the Characteristics and Intentions of Sharing Economy Services*

The consumer attitude and consumer trust are found to be mediated between the characteristics of sports sharing economy service and the intention to use it. In Li's study (2021), the sharing economy accommodation service was found to have a positive effect on the use intention through consumer attitude and consumer trust. In the research by Lee (2017) for the relationship between sharing economy service, car sharing experience, and consumer attitude, the characteristics of sharing was found to have a positive effect on the intention to use through consumer attitude. Therefore, the following hypotheses were established based on these theoretical grounds.

**Hypothesis 6 (H6).** *The characteristics of sports sharing economy services influence the use intention through the consumer attitude and consumer trust.*

**3. Methods**

This study was designed to verify the mediating effects between consumer attitude and trust level regarding the influence of characteristics in sports sharing economy services

on use intention. In other words, it analyzed the influence of the characteristics of the independent variables of sports sharing services, which influence the intention to use in the dependent variables, and also verified the mediating effects of the attitude and trust of consumers in a relationship in which the characteristics of sports sharing economy services influence the use intention.

### 3.1. Study Subjects

This study applied a random sampling method on colleges across four cities (Seoul, Gyeonggi-do, Gangwon-do, and Choongchung-do). Additionally, a survey was conducted among students in their first to fourth years of study. The researchers distributed 360 online survey questionnaires from 7 December to 20 December 2020, which were collected through email or as a Google survey. Of the 360 survey questionnaires, 302 were finally used in the data analysis, omitting the 58 copies with insincere answers (Table 1).

**Table 1.** Demographic characteristics of the subjects ($n = 302$).

|  | Category | Frequency | Percent |
|---|---|---|---|
| Gender | Man | 195 | 64.6 |
|  | Woman | 107 | 35.4 |
| University year | Freshman | 77 | 25.5 |
|  | Sophomore | 93 | 30.8 |
|  | Junior | 80 | 26.5 |
|  | Senior | 52 | 17.2 |
| Age | <21 | 73 | 24.2 |
|  | 21 to 22 | 92 | 30.5 |
|  | 23 to 24 | 85 | 28.1 |
|  | ≥25 | 52 | 17.2 |

### 3.2. Measurement Tool

The characteristics of sports sharing economy services served as the measurement tools in this study, and the sub-factors of this measurement tool included bonding, sustainability, usefulness, risk, consumer attitude, trust level, and use intention.

To evaluate the usefulness, this study presented five questions rated with a 5-point Likert scale by modifying the scale used by Cho (2019), which was based on the study by Venkatesh and Davis (2000), to fit sports sharing economy services. Usefulness was operationally defined as the degree of economic and effective benefits from using sharing economy services. Usefulness questionnaires included the following five questions: 'Using the sharing economy service is cheaper than purchasing'; 'Using the sharing economy service is efficient and useful'; 'Access to the sharing economy service is easy and convenient'; 'The sharing economic service has a high level of satisfaction (cost-effectiveness) for its price'; and 'The sharing economy service reduces the time and effort required to search for purchase information'.

The tool used to evaluate bonding involved four questions rated with a 5-point Likert scale by modifying the scale used by Cho (2019), which was based on the study by Summers et al. (2010) to fit the sports sharing economy services. In this study, bonding was operationally defined as a psychological sense of belonging that is mutually anticipated while using sharing economy services. Bonding questionnaires included the following four questions: 'I think other people who use sharing economy services have similar tendencies as me'; 'I think others who use the sharing economy service feel socially connected to me'; 'I think I can build a bond with other people who use the sharing economy service'; and 'I think I can make friends with other people who use the sharing economy service'.

Additionally, the measurement tool for sustainability was presented with four questions rated using a 5-point Likert scale by modifying the scale used by Cho (2019), which was based on the study by Hamari et al. (2016), to fit the sports sharing economy services. In this study, sustainability was operationally defined as saving resources without

causing environmental damage. Sustainability questionnaires included the following four questions: 'Sharing economy services are a sustainable consumption way that can solve environmental problems'; 'The sharing economy service is a consumption way that can reduce the waste of resources and energy'; 'The sharing economy service is an environmentally friendly consumption way that can protect the environment'; and 'Sustainable growth will be possible through sharing economy services'.

The evaluation for risk was conducted using a tool that included four questions rated using a 5-point Likert scale by modifying the scale used by Cho (2019), which was based on the study by Toni et al. (2018), to fit the sports sharing economy services. In this study, risk was operationally defined as the extent of concern that is felt from the negative consequences that are likely to occur when using sharing economy services. Risk questionnaires included the following four questions: 'There is concern that the service content presented online and the service content to be experienced may differ'; 'Public safety and personal safety are concerned in the process of using sharing economy services'; 'I am concerned about whether the content presented through the online contract will be kept as it is'; and 'I am concerned about whether compensation for losses incurred while using the service is properly compensated'.

The tool for measuring consumer attitude was six questions rated using a 5-point Likert scale by modifying the scale, which was used by Kang (2014) and modified by Ryu (2020), based on the study by Cho (2019) to fit the sports sharing economy services. Consumer attitude was operationally defined as the degree of favorable emotion regarding sharing economy services. Consumer attitude questionnaires included the following six questions: 'I think it is a good experience to use the sharing economy service'; 'I think it is good to use the sharing economy service'; 'I think it is wise to use the sharing economy service'; 'I think it is beneficial to use the sharing economy service', 'I think it is worthwhile to use the sharing economy service'; and 'I think that using the sharing economy service is to form a cooperative relationship with others'.

The measurement tool for the trust level was a modified scale containing six questions rated using a 5-point Likert scale to fit the environment of sharing economy services based on the study by Koufaris and Hampton-Sosa (2004). Trust was operationally defined as the degree to which consumers believe in sharing economy services. Trust questionnaires included the following six questions: 'Sharing economy services will provide reliable services'; 'The sharing economy service will provide a great service'; 'The information provided by the sharing economy service is reliable'; 'The sharing economy service will provide the prescribed service as promised', 'The sharing economy service will provide a service that eliminates risk factors'; and 'I think appropriate compensation will be provided when services are not provided as promised'.

Use intention was operationally defined as the degree to which it influences future behavior to use the sharing economy services in advance. This was evaluated through four questions rated using a 5-point Likert scale that modified and supplemented the scale used by Lee (2020), which was based on the study to fit the sports domain. Use intention questionnaires included the following four questions: 'When I need to use the service, I will use the sharing economy service'; 'I am willing to recommend people around me to use the sharing economy service'; 'I will actively use sharing economy services in the future'; and 'I have plans to use sharing economy service products in various fields'.

### 3.3. Data Analysis

This study used the SPSS ver. 25.0 program for the statistical analysis of data. The value of Cronbach's $\alpha$ was calculated to verify the reliability of the scale, and descriptive statistics were used to determine the general characteristics of study subjects and the characteristics of major variables. The correlation coefficient was calculated to determine the relationship between major variables such as the characteristics of sports sharing economy services, attitude, degree, and use intention. To verify the mediated effect, verification was conducted using the method proposed by Baron and Kenny (1986).

*3.4. Validity and Reliability of the Measurement Tools*

The validity of this study complied with the criterion that a factor value over 0.60 is acceptable in the empirical data that was argued by Spearman (1904) and suggested in the exploratory factor analysis by Lee et al. (2016a,2016b). The reliability was verified with a criterion that Cronbach's α has an internal consistency ranging from 0.957 to 0.853 (Nunnally and Bernstein 1994). The validity and reliability of the aforementioned measurement tools were confirmed to fall within abovementioned criteria.

The results of an exploratory factor analysis on the characteristics of sports sharing economy services showed that four factors were obtained from a total of 17 questions. Factor 1 included "risk" with an eigenvalue of 3.564 and description variance of 20.965%; factor 2 was "usefulness" with an eigenvalue of 3.210 and description variance of 18.880%; factor 3 was "sustainability" with an eigenvalue of 2.795 and description variance of 16.441%%; factor 4 was "bonding" with an eigenvalue of 2.784 and description variance of 16.376%. These showed a factor value of 0.944~0.715, indicating that the validity was secured. Additionally, as for the reliability, Cronbach's α showed 0.957~0.853, and was considered to have internal consistency (Nunnally and Bernstein 1994).

The results of the factor analysis on the consumer attitude of sports sharing economy services showed that one factor was extracted from a total of six questions; the eigenvalue was 4.119, the description variance was 68.655%, and the reliability was 0.908.

The results of the factor analysis on the level of consumer trust in the sports sharing economy services showed that one factor was obtained from six questions: the eigenvalue was 3.717, the description variance was 61.954%, and the reliability was 0.877.

The results of examining the factor analysis on the degree of use intention in the sports sharing economy services indicated that one factor was extracted from six questions; the eigenvalue was 3.066, the description variance was 76.639%, and the reliability was 0.898.

## 4. Results

### 4.1. Correlation Analysis

The results of correlations between the four factors showed that the characteristics of sports sharing economy services exhibited a significant correlation with consumer attitude, trust, and use intention. The usefulness of sports sharing economy services was significantly correlated with bonding ($r = 0.369$, $p < 0.01$), sustainability, and consumer attitude ($r = 0.388$, $p < 0.01$). The trust level and use intention showed the most significant correlations with consumer attitude ($r = 0.436$, $p < 0.01$) and sustainability ($r = 0.458$, $p < 0.01$), respectively.

### 4.2. Study Hypothesis Test

4.2.1. Influence of the Characteristics of Sharing Economy Services on the Use Intention

In this study, a regression analysis was conducted on the influence of the characteristics of sports sharing economy services on use intention. The analysis results showed a statistical significance, indicating a coefficient of determination $R^2 = 0.369$ (F = 43.447 ($p < 0.001$)). The factors of usefulness (β = 228, $p < 0.001$), bonding (β = 179, $p < 0.001$), and sustainability (β = 332, $p < 0.001$) of the sports sharing economy services exhibited a positive (+) influence on the use intention, whereas the risk (β = −199, $p < 0.001$) factors in the characteristics of sports sharing economy services showed a negative (−) influence.

4.2.2. Mediating Effect of Consumer Attitude on the Characteristics and Intentions of Sharing Economy Services

Table 2 shows the results of conducting the mediating regression analysis by Baron and Kenny (1986) to determine whether consumer attitude plays a mediating role in the characteristics of sports sharing economy services under H2. The hypothesis was adopted as the characteristic of all sub-factors that were validated as significant. To determine whether consumer attitude plays a mediating role in the relationship between usefulness and use intention of the sports sharing economy services, the influence of usefulness in sports sharing economy services on consumer attitude was analyzed in the first stage.

Consequently, the explanatory power of the regression equation in the sample was 13.9% ($R^2$ = 0.139), and the regression model (F = 48.546, $p < 0.001$) was found to be significant. In the second stage, the influence of usefulness of sports sharing economy services on use intention was analyzed. The explanatory power of the regression equation in the sample was 16.9% ($R^2$ = 0.169), and the regression model (F = 60.960, $p < 0.001$) was found to be significant. In the third stage, the influence of both usefulness of the sports sharing economy services and consumer attitude on use intention were analyzed. The explanatory power of the regression equation in the sample was 24.8% ($R^2$ = 0.248), and the regression model (F = 49.277, $p < 0.001$) was found to be significant; therefore, H2a was supported.

**Table 2.** Mediating effect of consumer attitude on the characteristics and intentions of sharing economy services.

| Step | Variable | B | SEB | β | F | $R^2$ |
|------|----------|------|------|-------|---------|----------|
| 1 | Usefulness → Consumer Attitude | 0.425 | 0.061 | 0.373 *** | 480.546 | 0.139 *** |
| 2 | Usefulness → Use Intention | 0.463 | 0.059 | 0.411 *** | 600.960 | 0.169 *** |
| 3 | Usefulness → Use Intention | 0.336 | 0.061 | 0.298 *** | 490.277 | 0.248 *** |
|   | Consumer Attitude → Use Intention | 0.300 | 0.054 | 0.303 *** |   |   |
| 1 | Bonding → Consumer Attitude | 0.417 | 0.061 | 0.365 *** | 160.146 | 0.133 *** |
| 2 | Bonding → Use Intention | 0.418 | 0.061 | 0.369 *** | 470.425 | 0.137 *** |
| 3 | Bonding → Use Intention | 0.285 | 0.062 | 0.252 *** | 430.779 | 0.227 *** |
|   | Consumer Attitude → Use Intention | 0.319 | 0.054 | 0.322 *** |   |   |
| 1 | Sustainability → Consumer Attitude | 0.456 | 0.063 | 0.388 *** | 530.055 | 0.150 *** |
| 2 | Sustainability → Use Intention | 0.533 | 0.060 | 0.458 *** | 790.685 | 0.210 *** |
| 3 | Sustainability → Use Intention | 0.408 | 0.062 | 0.350 *** | 560.916 | 0.276 *** |
|   | Consumer Attitude → Use Intention | 0.276 | 0.053 | 0.278 *** |   |   |
| 1 | Risk → Consumer Attitude | −0.151 | 0.033 | −0.257 *** | 210.237 | 0.066 *** |
| 2 | Risk → Use Intention | −0.168 | 0.032 | −0.288 *** | 270.215 | 0.083 *** |
| 3 | Risk → Use Intention | −0.113 | 0.031 | −0.195 *** | 390.016 | 0.207 *** |
|   | Consumer Attitude → Use Intention | 0.361 | 0.053 | 0.364 *** |   |   |

*** $p < 0.001$.

To determine whether consumer attitude plays a mediating role in the relationship between bonding and the intention to use sports sharing economy services, the first stage analyzed the influence of the bonding of sports sharing economy services, which is an independent variable, on consumer attitude, which is a parameter. Consequently, the explanatory power of the regression equation in the sample was 13.3% ($R^2$ = 0.133), and the regression model (F = 16.146, $p < 0.001$) was found to be significant. The second stage analyzed the influence of bonding in sports sharing economy services on use intention. The explanatory power of the regression equation in the sample was 13.7% ($R^2$ = 0.137), and the regression model (F = 47.425, $p < 0.001$) was found to be significant. The third stage analyzed the influence of both bonding and consumer attitude on the use intention. The explanatory power of the regression equation in the sample was 22.7% ($R^2$ = 0.227), and the regression model (F= 43.779, $p < 0.001$) was found to be significant; therefore, H2b was supported.

To determine whether consumer attitude plays a mediating role in the relationship between sustainability and the use intention of sports sharing economy services, the first stage analyzed the influence of sustainability in a sports sharing economy service on consumer attitude. Consequently, the explanatory power of the regression equation in the sample was 15.0% ($R^2$ = 0.150), and the regression model (F = 53.055, $p < 0.001$) was found to be significant. The second stage analyzed the influence of sustainability in sports sharing economy services on use intention. The explanatory power of the regression equation in the sample was 21.0% ($\chi^2$ = 0.210), and the regression model (F = 79.685, $p < 0.001$) was found to be significant. The third stage analyzed the influence of both sustainability and consumer attitude on the use intention. The explanatory power of the regression equation

in the sample was 27.6% ($R^2 = 0.276$), and the regression model (F = 56.916, $p < 0.001$) was found to be significant; therefore, H2c was supported.

To determine whether consumer attitude plays a mediating role in the relationship between risk and the use intention of the sports sharing economy services, the first stage analyzed the influence of the risk in sports sharing economy services on consumer attitude. The explanatory power of the regression equation in the sample was 6.6% ($R^2 = 0.066$), and the regression model (F = 21.237, $p < 0.001$) was found to be significant. The second stage analyzed the influence of risk in sports sharing economy services on use intention. The explanatory power of the regression equation in the sample was 8.3% ($R^2 = 0.083$), and the regression model (F = 27.215, $p < 0.001$) was found to be significant. The third stage analyzed the influence of both risk and consumer attitude on the use intention. The explanatory power of the regression equation in the sample was 20.7% ($R^2 = 0.207$), and the regression model (F = 39.016, $p < 0.001$) was found to be significant; therefore, H2d was supported.

### 4.2.3. Mediating Effect of Trust Level on the Characteristics and Intentions of Sharing Economy Services

As shown in Table 3, the mediating effect of a trust level on the characteristics and use intention of the sports sharing economy services appeared to influence each of the sub-factors; therefore, H3 was adopted accordingly. This is the result of testing the hypothesis on trust regarding the usefulness and use intention among the characteristics of the sports sharing economy services. In the first stage, the influence of usefulness of the sports sharing economy services on the trust level was analyzed. Consequently, the explanatory power of the regression equation in the sample was 16.5% ($R^2 = 0.165$), and the regression model (F = 59.395, $p < 0.001$) was found to be significant. The second stage analyzed the influence of usefulness in sports sharing economy services on use intention. The explanatory power of the regression equation in the sample was 16.9% ($R^2 = 0.169$), and the regression model (F = 60.960, $p < 0.001$) was found to be significant. The third stage analyzed the influence of both usefulness and reliability level on the use intention. The explanatory power of the regression equation in the sample was 28.7% ($R^2 = 0.287$), and the regression model (F = 60.106, $p < 0.001$) was found to be significant; therefore, H3a was supported.

**Table 3.** Mediating effect of consumer's trust on the characteristics and intentions of sharing economy services.

| Step | Variable | B | SEB | β | F | $R^2$ |
|---|---|---|---|---|---|---|
| 1 | Usefulness → Consumer Trust | 0.434 | 0.056 | 0.407 *** | 590.395 | 0.165 *** |
| 2 | Usefulness → Use Intention | 0.463 | 0.059 | 0.411 *** | 600.960 | 0.169 *** |
| 3 | Usefulness → Use Intention | 0.291 | 0.060 | 0.258 *** | 600.106 | 0.287 *** |
|  | Consumer Trust → Use Intention | 0.397 | 0.057 | 0.376 *** |  |  |
| 1 | Bonding → Consumer Trust | 0.466 | 0.056 | 0.436 *** | 700.432 | 0.190 *** |
| 2 | Bonding → Use Intention | 0.418 | 0.061 | 0.369 *** | 470.425 | 0.137 *** |
| 3 | Bonding → Use Intention | 0.223 | 0.062 | 0.197 *** | 530.256 | 0.263 *** |
|  | Consumer Trust → Use Intention | 0.417 | 0.058 | 0.395 *** |  |  |
| 1 | Sustainability → Consumer Trust | 0.463 | 0.058 | 0.420 *** | 640.429 | 0.177 *** |
| 2 | Sustainability → Use Intention | 0.533 | 0.060 | 0.458 *** | 790.685 | 0.210 *** |
| 3 | Sustainability → Use Intention | 0.362 | 0.062 | 0.311 *** | 670.391 | 0.311 *** |
|  | Consumer Trust → Use Intention | 0.379 | 0.056 | 0.350 *** |  |  |
| 1 | Risk → Consumer Trust | −0.147 | 0.031 | −0.268 *** | 230.140 | 0.072 *** |
| 2 | Risk → Use Intention | −0.168 | 0.032 | −0.288 *** | 270.215 | 0.083 *** |
| 3 | Risk → Use Intention | −0.100 | 0.030 | −0.172 ** | 520.146 | 0.259 *** |
|  | Consumer Trust → Use Intention | 0.460 | 0.055 | 0.435 *** |  |  |

** $p < 0.01$, *** $p < 0.001$.

In the first stage, the influence of bonding in the sports sharing economy on the reliability level was analyzed. From the results, the explanatory power of the regression equation in the sample was 19.0% ($R^2 = 0.190$), and the regression model (F = 70.432, $p < 0.001$) was found to be significant. The second stage analyzed the influence of bonding in sports sharing economy services on use intention. The explanatory power of the regression equation in the sample was 13.7% ($R^2 = 0.137$), and the regression model (F = 47.425, $p < 0.001$) was found to be significant. The third stage analyzed the influence of both bonding and the reliability level on the use intention. The explanatory power of the regression equation in the sample was 26.3% ($R^2 = 0.263$), and the regression model (F = 53.256, $p < 0.001$) was found to be significant; therefore, H3b was supported.

In the first stage, the influence of sustainability of the sports sharing economy services on consumer attitude was analyzed. From the results, the explanatory power of the regression equation in the sample was 17.7% ($R^2 = 0.177$) and the regression model (F = 64.429, $p < 0.001$) was found to be significant. The second stage analyzed the influence of the sustainability of sports sharing economy services on use intention. The explanatory power of the regression equation in the sample was 21.0% ($\chi^2 = 0.210$), and the regression model (F = 79.685, $p < 0.001$) was found to be significant. The third stage analyzed the influence of both sustainability and consumer attitude on the use intention. The explanatory power of the regression equation in the sample was 31.1% ($R^2 = 0.311$), and the regression model (F = 67.391, $p < 0.001$) was found to be significant; therefore, H3c was supported.

To determine whether reliability level plays a mediating role in the relationship between risk and the use intention of the sports sharing economy services, a mediation regression analysis was conducted. The analysis results were as follows. In the first stage, the influence of risk in sports sharing economy services on consumer attitude was analyzed. The explanatory power of the regression equation in the sample was 7.2% ($R^2 = 0.072$), and the regression model (F = 23.140, $p < 0.001$) was found to be significant. The second stage analyzed the influence of risk in sports sharing economy services on use intention. The explanatory power of the regression equation in the sample was 8.3% ($R^2 = 0.083$), and the regression model (F = 27.215, $p < 0.001$) was found to be significant. The third stage analyzed the influence of both risk and the reliability level on the use intention. The explanatory power of the regression equation in the sample was 25.9% ($R^2 = 0.259$), and the regression model (F = 52.146, $p < 0.001$) was found to be significant; therefore, H3d was supported.

### 4.2.4. Influence on the Consumer Attitude of Sharing Economy Services and the Use Intention

As a result of verifying the relationship between consumer attitude and the use intention of the sharing economy services, a regression analysis was conducted to determine whether the consumer attitude toward sports sharing economy services influences the use intention. Additionally, the coefficient of determination was $R^2 = 0.172$, F = 62.114, which was statistically significant ($p < 0.001$). The trust level (ß = 0.414, $p < 0.001$) factor in sports sharing economy services was shown to have a positive influence on use intention; therefore, H4 was supported. Furthermore, as a result of verifying the relationship between trust level and use intention in sharing economy services, a regression analysis was conducted to determine whether the trust level in sports sharing economy services influences use intention. The coefficient of determination was $R^2 = 0.231$, F = 90.175, which was statistically significant ($p < 0.001$). The trust level (ß = 0.481, $p < 0.001$) factor in sports sharing economy services was shown to have a positive influence on use intention; therefore, H5 was supported.

### 5. Discussion

This study conducted an empirical analysis on whether consumer attitude and trust level play a mediating role in the relationship between the characteristics of sports sharing economy services and the use intention of consumers. There were three research questions

in the study. The first question was whether the characteristics of sports sharing economy services affect the consumers' use intention. The second question was whether the characteristics of sports sharing economy services affect the attitude of consumers and the trust level in sports sharing economy services. The third question was whether the attitude and trust level of consumers mediate sports sharing economy services and use intention. The test results of the hypotheses derived according to the purposes of the study revealed the following.

First, the characteristics of sports sharing economy services influence purchase intention. Usefulness, bonding, and sustainability had a positive influence, while risk had a negative influence. These results are consistent with the outcomes of a study that was conducted using the motivation theory and TAM, to evaluate the relationship between the characteristics and the use intention of the sharing economy services (Shin and Han 2019). These results are also consistent with the outcomes of a study that showed that the perceived risk of consumers has a negative influence on purchase intention (Joo et al. 2008; Hong 2004). Moreover, these results are partially consistent with the study conducted by Choi (2019), who reported that the sharing economy is an important variable to the purchase intention of consumers. Another study by Cho (2019), stated that usefulness, bonding, and sustainability among the characteristics of the sharing economy services have a positive influence on purchase intention. These can be deemed consistent with the results owing to the fact that the characteristics of sports sharing economy services provide new experiences that cannot be delivered by the existing economy system. Moreover, the concept of the services is sold based on the experiences, which are digital contents distributed through the Internet (Lee et al. 2016b). Therefore, the application to activate the use of bikes and electric kickboards in consideration of the characteristics of sports sharing economy services should be upgraded more easily and conveniently. A system that can complete the purchase in a no-touch state should be established before a consumer uses the device. Additionally, from the perspective of reducing the risks of consumers, a stable purchasing system should be implemented for the continuous purchase of the service. Moreover, the production of antibacterial materials for sports equipment should also be considered.

Second, the characteristics of sports sharing economy services have an influence on consumer attitude. This is consistent with the findings by Abadi et al. (2012) and Hamari et al. (2016). Consumers mostly exhibited a positive attitude toward sharing economy services. Such results are mainly attributable to the fact that people in their 20s are more familiar with the payment method in sharing economy services than other age groups, and bike and electric kickboards are the only categories in sports that can be enjoyed under the COVID-19 situation, making it easier for college students to use them as an economic means of facilitating short-distance movement. This tendency is expected to continue even after the COVID-19 pandemic; therefore, it is necessary to expand them to individual sports rather than group sports. In other words, it is necessary to develop not only outdoor sports, but also indoor sports so as to enable home training. Moreover, it is necessary to establish a means to earn profit and expand the scope of sharing by applying them to the sharing economy services (Kim 2019).

Third, the characteristics of sports sharing economy services influence the trust level of consumers. This is consistent with the results of Shim (2016) and Lee and Noh (2008). Additionally, this study is supported by the argument of Hao and Kim (2020), who stated that the characteristics of sharing economy services influence the trust stemming from the internal and external motivations of consumers. Further, risk can be considered the most important characteristic of the trust of consumers with regard to sports sharing economy services. Therefore, to expand the sports sharing economy services, the relevant service providers need measures to lower the damage rate of electric kickboards, consider an efficient charging system, and reduce risks to both users and pedestrians by strictly applying the safety regulations when operating the service during the day and at night.

Fourth, consumer attitude with regard to the sports sharing economy services was found to have a positive influence on the use intention, a result that was proven using the rational behavioral theory. Moreover, the result is also consistent with many studies (Song and Huh 2020; Yang et al. 2020; Zhang and Ryu 2020) which state that a favorable consumer attitude has a positive influence on use intention. Meanwhile, consumer attitude has an influence on the socio-demographic characteristics and reference group types (Lee 2016). Therefore, in addition to sharing bikes and electric kickboards that are mainly used by people in their 20s and 30s, the strategies that enable individuals and families to enjoy the services safely should be sought by developing various categories that focus on individual sport events as sports sharing economy services.

Fifth, the mediating effect was verified because the trust of sports sharing economy services had a positive influence on use intention. Consumer attitude with regard to sports sharing economy services has a positive influence on use intention. This result is consistent with previous studies (Hao and Kim 2020), which state that the use intention increases as the attitude toward the sports sharing economy services becomes favorable. Currently, South Korea uses reliable overseas sports sharing economy services. Such sports sharing economy services can be expanded only under the trust of consumers. Therefore, it is necessary to establish the trust of consumers with regard to online home training by introducing overseas hub services such as Lime, Uber, MIRROR, and PELOTON, which are considered as sports-specific economy services in the US. To activate the sports industry, it is necessary to create a model that enables interaction as a Korean-style sustainable sports sharing economy service.

## 6. Conclusions

This study is significant in that it elucidates that the characteristics of sharing economy service in the sports domain influence not only the consumer attitude on use intention and trust level, but also use intention. However, this study has some limitations, including the following.

First, it is difficult to generalize the findings of this study to those of other countries or all age groups because the participants were South Korean college students. In future studies, it is necessary to investigate the relationships among the characteristics, use intentions, consumer attitude, and trust levels in sports sharing economy services for consumers in various countries and age groups.

Second, this study selected consumer attitude and trust level as parameters between the sports sharing economy and use intention based on previous studies. In the future, studies could be conducted using moderating variables such as age and economic level, which can better demonstrate the relationship between the characteristics of sports sharing economy services and use intention.

Third, this study has meaning as a study that verified the mediating effect of trust factors in the relationship between the characteristics of sharing economy services and the use intention. As can be seen from the car-sharing service study by Cho (2019), however, the trust factor can moderate the characteristics of the sharing economy service and the behavior of consumers. Therefore, in a follow-up study, it is recommended to conduct a study that verified the trust factor as a moderating variable in the relationship between the characteristics of the sports sharing economy service and the intention to use it.

**Author Contributions:** S.K. made substantial contributions to drafting of manuscript and study conception and design. K.L. conducted analysis on all samples, interpreted data and acted as corresponding author. S.K. and K.L. collected data and helped to interpret data and evaluate and edit the manuscript. All authors have read and approved the final version submitted for publication, and agree to toe order in which the authors are listed. All authors have read and agreed to the published version of the manuscript.

**Funding:** This research received no external funding.

**Conflicts of Interest:** The authors declare no conflict of interest.

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
