# Peer review of "Influence of the Characteristics of Sports Sharing Economy Services on Intention of Use: The Mediating Effect of Consumer Attitude and Trust"

_socsci, doi:10.3390/socsci11020057_

Round 1

Reviewer 1 Report

Thank you for giving me the possibility to review this paper. The paper is of high quality, will be interesting to readers who are dealing with the issues of sharing economy and consumer behavior.

The abstract is well structured. The keywords are in line with the terms used in the research.  Arguments and discussion of findings are coherent, balanced, and compelling. The references are correct of which most are up to date, and those from previous decades are relevant. The English language and style are fine, from my point of view.

As to paper structure, the Introduction section correctly puts the research topic in context and the wording is appropriate and meaningful. Still, this section is not in the traditional way, with the objectives of the article and background to the paper provided, and the structure of the paper presented in the last paragraph. I am not sure that the authors should give study hypotheses in this part. I also would prefer to see a separate section presenting a literature review.

The methodology is appropriate, relevant to understanding the phenomenon under investigation. It makes sense to use the theoretical lens of Veblen's social theory of consumption, about bonding capital and its influence on intentions. The technique is also appropriate - mediating regression analysis (from Baron, Daivd, 1986). I have not met before any paper with mediating regression analysis applied to the issues discussed in the paper.

The authors include important information about the empirical research they have carried. The sources of empirical information, the sample, and the date are correctly detailed. Maybe it could be interesting for readers to see also info about the survey itself (questionnaire, formulations, nor response frequencies), added to the most important final tables presented in the paper. But it is up to the authors since the presented info is enough to adequately understand the results.

It will be better to divide the final part into two parts, namely Discussion and Conclusion sections. In the last (but not least) one, the authors could more thoroughly present their statement on contributions to the literature, practical implications, limitations of the research, as well as future research paths.

Overall, the paper is recommended for publication, with minor revisions proposed above.

Reviewer 2 Report

Overall, I find this study with a good level of relevance and pertinence to fulfill certain gaps in the literature and advance the body knowledge, I would say, about the Sharing Economy and, more particularly, about understanding more clearly the reasons, motivations, what might influence consumers/users to engage in Sharing Economy services. Congratulations on that, dear authors.

However, at this moment, I just have two comments and suggestions that might help improve the overall quality of the manuscript:

  1. Please state a clear Research Question (RQ) both in the Abstract and along with the text (example: integrated at the end of the Introduction section, beginning of the Discussion section, and beginning of the Conclusion section);
  2. I have my doubts that the variable of "Trust" plays a role as a mediator and not as a moderator. Classically, "Trust" should play a role as a moderator (meaning: it may intensify or lessen a given intention to incur in the use, adherence, purchase, etc. of something). I suggest that the authors, should address this possibility, at least theoretically (particularly, under the 1.1 Study Hypotheses subsection). I also suggest the authors to read more about other studies regarding the reasons, antecedents, motivations for peer-to-peer and business-to-peer participation, and thus, enrich their Introduction section with further literature. Examples that might be helpful to the authors:
      1. Hawlitschek, Teubner & Weinhardt (2016) Trust in the Sharing Economy;
      2.  
      3. Mohlmann & Geissinger (2018) Trust in the Sharing Economy: Platform-Mediated Peer Trust;
      4.  
      5. Benoit et al. (2017) A triadic framework for collaborative consumption (CC): Motives, activities and resources & capabilities of actors;
      6.  
      7. Godelnik (2017) Millennials and the sharing economy: Lessons from a ‘buy nothing new, share everything month’ project;
      1. Mugion et al. (2019) Understanding the antecedents of car-sharing usage. An empirical study in Italy;
      2.  
      3. Bocker and Meelen (2017) Sharing for people, planet or profit? Analysing motivations for intended sharing economy participation;
      4.  
      5. Hawlitschek, Teubner & Gimpel (2018) Consumer motives for peer-to-peer sharing;
      1. Mittendorf (2017) The Implications of Trust in the Sharing Economy – An Empirical Analysis of Uber;
      2.  
      3. Hawlitschek, Teubner & Gimpel (2018) Consumer motives for peer-to-peer sharing;

I wish the authors the best in their revision and advancing with the publication of this study.

Thank you very much for your contribution for the the body of knowledge of the Sharing Economy.

Academic regards.
